# Discriminating cross-reactivity in polyclonal IgG1 responses against SARS-CoV-2 variants of concern

Danique M. H. van Rijswijck [1,2,6], Albert Bondt [1,2,6], Max Hoek[1,2], Karlijn van der Straten[3,4], Tom G. Caniels [3], Meliawati Poniman[3], Dirk Eggink[5], Chantal Reusken [5], Godelieve J. de Bree[4], Rogier W. Sanders [3], Marit J. van Gils [3] & Albert J. R. Heck [1,2] ✉

Existing assays to measure antibody cross-reactivity against different SARS-CoV-2 spike (S) protein variants lack the discriminatory power to provide insights at the level of individual clones. Using a mass spectrometry-based approach we are able to monitor individual donors' IgG1 clonal responses following a SARS-CoV-2 infection. We monitor the plasma clonal IgG1 profiles of 8 donors who had experienced an infection by either the wild type Wuhan Hu-1 virus or one of 3 VOCs (Alpha, Beta and Gamma). In these donors we chart the full plasma IgG1 repertoires as well as the IgG1 repertoires targeting the SARS-CoV-2 spike protein trimer VOC antigens. The plasma of each donor contains numerous anti-spike IgG1 antibodies, accounting for <0.1% up to almost 10% of all IgG1s. Some of these antibodies are VOC-specific whereas others do recognize multiple or even all VOCs. We show that in these polyclonal responses, each clone exhibits a distinct cross-reactivity and also distinct virus neutralization capacity. These observations support the need for a more personalized look at the antibody clonal responses to infectious diseases.

Since the outbreak, the SARS-CoV-2 virus has spread rapidly around the world and is continuously presenting itself in new variants[1]. While most mutations are mildly deleterious, certain mutations lead to variants with altered virus characteristics, affecting transmissibility and antigenicity[1]. Variants that affect virus characteristics and that cause significant community transmission have been declared Variants of Concern (VOC)[2–4]. Currently, five mutated SARS-CoV-2 VOCs have been annotated by the WHO: Alpha (B.1.1.7), Beta (B.1.351), Gamma (B.1.1.28.P1), Delta (B.1.617.2) and Omicron (B.1.1.529), although the latter seems to be leading to less severe disease[2]. Since the virus has

spread globally, and also due to intense worldwide vaccination programs, a substantial amount of the population has created an immune response and subsequent memory towards the virus[5].

However, with the emergence of new VOCs, the question becomes relevant whether the humoral immunity that is gained after infection with one VOC also provides protection against another VOC. This cross-reactive protection against different variants would be essential to ultimately combat the virus. Additionally, it may be interesting to see whether there are differences in response and cross-reactivity between individuals, in between different antigen directed Ig

[1]Biomolecular Mass Spectrometry and Proteomics, Bijvoet Center for Biomolecular Research and Utrecht Institute for Pharmaceutical Sciences, University of Utrecht, Padualaan 8, Utrecht 3584 CH, The Netherlands. [2]Netherlands Proteomic Center, Padualaan 8, Utrecht 3584 CH, The Netherlands. [3]Department of Medical Microbiology, Amsterdam UMC, University of Amsterdam, Amsterdam Institute for Infection and Immunity, Meibergdreef 9, Amsterdam 1105 AZ, The Netherlands. [4]Department of Internal Medicine, Amsterdam UMC, Vrije Universiteit Amsterdam, Amsterdam Institute for Infection and Immunity, Meibergdreef 9, Amsterdam 1105 AZ, The Netherlands. [5]National Institute for Public Health and the Environment, RIVM, Antonie van Leeuwenhoeklaan 9, 3721 MA Bilthoven, The Netherlands. [6]These authors contributed equally: Danique M.H. van Rijswijck, Albert Bondt. ✉e-mail: a.j.r.heck@uu.nl

clones, and in between SARS-CoV-2 variants. Previous reports on this topic have suggested that new variants can (partially) escape humoral immune responses, since no clear binding or neutralization was observed in vitro[4]. However, in vitro assays do not provide the full picture of the immune response and in vivo studies are needed to support this further. Moreover, the currently available assays used to assess cross-reactivity against different VOCs, target the response at the level of the total antibody pool, overlooking the fact that our immune system provides a polyclonal response creating multiple different antibody clones against a single antigen. In principle, all these different clones would give a unique response and can therefore show broad variation in the degree of cross-reactivity towards the antigens of other VOCs. Recently, there have been advances in techniques for antibody repertoire profiling, such as next-generation sequencing (NGS) of B cells. However, this technique focuses on the circulating B cell population which might not represent the antibodies in circulation. Furthermore, it does not provide information regarding the abundances of each clone that will eventually end up in the circulation[6]. This defines the unmet need for an approach that provides information on individual antibody clones in the circulation to better understand the elicited antibody response after infection. Subsequently, this information will be crucial for the development of optimal biotherapeutics, that ideally are cross-reactive and neutralizing against all known and/or future VOCs.

Previously, our laboratory introduced an approach to monitor qualitatively and quantitatively clonal IgG1 repertoires in plasma, allowing investigation of humoral immunity at the molecular level in detail. In this approach IgGs are purified and subsequently cleaved into the constant domain (Fc) and antigen binding domain (Fab), using an enzyme that cleaves only IgG1s. All released IgG1 Fab molecules, spanning typically a 45 kDa <Mw <53 kDa mass range, are then fractionated and profiled at the intact protein level by liquid-chromatography coupled mass spectrometry (LC−MS). As each clone has a distinct mass and retention time (RT), these LC−MS traces provide a qualitative picture of the IgG1 repertoires. By spiking in recombinant IgG1 mAbs as internal standards each individual plasma clone in the plasma can be quantified[7]. In our initial work we focused on total IgG1 repertoires in plasma. Here, we extend this approach, not only monitoring total plasma clonal repertoires, but also we employed recombinant SARS-CoV-2 (2 P stabilized) spike trimer-proteins (in this manuscript referred to as S-protein) to enrich for the clonal repertoires targeting specifically SARS-CoV-2 S-protein antigens of the studied VOCs. This approach allows us to examine cross-reactivity of specific antibody clones against different S-protein variants.

In this work, we monitor the SARS-CoV-2 S-protein specific IgG1 polyclonal response in 8 selected donors that had suffered an infection with different SARS-CoV-2 VOCs, in total two donors (one male/one female) per variant. We assess the antibody binding towards the wild type Wuhan Hu-1 (WT) and three of the VOCs of the SARS-CoV-2 S-protein namely, Alpha, Beta, and Gamma. Our data show that the immune system produces in each donor a unique polyclonal IgG1 repertoire against the S-protein. Testing the cross-reactivity of each of the detected clones against other VOCs, we observe a broad spectrum of different responses. Some clones bind equally well to all VOCs S-proteins, whereas others only to one, two or three of the variants.

## Results
### IgG1 clonal profiling
The primary aim of this study was to assess the clonal diversity of SARS-CoV-2 specific IgG1 antibodies in plasma of individual donors and their cross-reactivity against the different S-protein variants representing VOCs. To validate the experimental approach used here we performed positive (anti- spike mAb binding) and negative control (non-anti-spike mAb binding and 'bare' beads binding) tests. From

these validation experiments we concluded that the method used here was highly specific both in terms of no non-specific binding towards the NHS agarose beads as well as specific Ig binding towards the S-protein (Supplementary Figs. 1 and 2). Moreover, previously we already showed that the here presented MS-based method results in highly reproducible data on IgG1 repertoires, both in biological and analytical replicates[7].

We analyzed the plasma of eight donors who had experienced an infection with the WT or one of the three VOCs, namely Alpha, Beta and Gamma (Fig. 1a, Supplementary Table 1). All plasma samples were collected 3−6 weeks after the start of symptoms. In total we recorded 40 different plasma IgG1 repertoires using LC−MS, of which 8 were full plasma repertoires (one per donor) and 32 were obtained from the S-protein variant directed sub-pools of IgG1s (Fig. 1b). Across all donors the number of detected IgG1 clones in full plasma as defined by LC−MS varied between 247−517, with the top 25 clones representing about 40−70% of the total IgG1 concentration (Supplementary Table 2). Focusing next on the Fabs retrieved by the immune-affinity pull-downs, it became noticeable that their total concentration was quite variable, with some donors producing anti S-protein IgG1s just up to a concentration of a few ng/mL (e.g., donor 002 and 307), whereas others produced up to >20 μg/mL of anti S-protein IgG1s (e.g., donor 003 and 303). This implies that in the former two donors the anti S-protein IgG1s make up less than 1% of the full plasma IgG1 repertoire, whereas in the latter case this is more in the range of 5−10%. The number of unique clones identified in all repertoires is provided in Supplementary Table 2 and shows that the total numbers align well with the total concentrations

Except donor 303, the highest number and the highest concentration of anti S-protein IgG1s was observed, as expected, versus the S-protein originating from the VOC causing the infection. Somewhat surprising, donor 303 was infected with the Alpha variant but displayed a substantially higher quantity of anti S-protein IgG1s in the WT pull-down when compared to the pull-down with the Alpha variant S-protein (Supplementary Table 2), despite the high similarity between Alpha and WT[4,8].

We next evaluated the diversity in these IgG1 repertoires intra- and inter-donors. Every unique IgG1 clone, annotated as $^{RT}\#_{mass}$, was distinguished by its accurate mass in Dalton and LC retention time (RT) in minutes. Adopting similar analysis tools as introduced earlier[7], we quantitatively overlapped the full plasma IgG1 repertoires, using the concentration of each unique identifier $^{RT}\#_{mass}$, whereby two identical profiles would provide an overlap of 100% (colored dark red in Fig. 2) and a weak overlap would be just a few % (colored white in Fig. 2). The observation that there is nearly no overlap in IgG1 repertoires in between donors is fully in line with our earlier data; each person's plasma IgG1 repertoire is unique and no clone is detected in more than one donor[7].

Next, we quantitatively overlapped the four affinity pulled-down repertoires for a single donor and compared that with the full plasma IgG1 repertoire of that given donor. Several interesting observations can be extracted from this data. For some donors (e.g. 003, 303, 304 and 310) the S-protein directed IgG1 repertoires are within a single donor quite alike, no matter which S-protein had been used as affinity handle. For donor 003 these four S-protein directed IgG1 repertoires also overlap quite well with the total IgG1 repertoire, indicating that several abundant clones in the plasma of this donor are directed against the S-protein of SARS-CoV-2 (Fig. 2). In contrast, with donor 002 the overlap among the four pull-downs is quite high, but the overlap with the full plasma IgG1 repertoire of that donor is close to zero, indicating that the majority of abundant clones found in the plasma of this donor are not directed against any of the four S-proteins.

Overall, we observe that in these repertoires quite a few clones are present that cross-react with other S-protein variants, although other

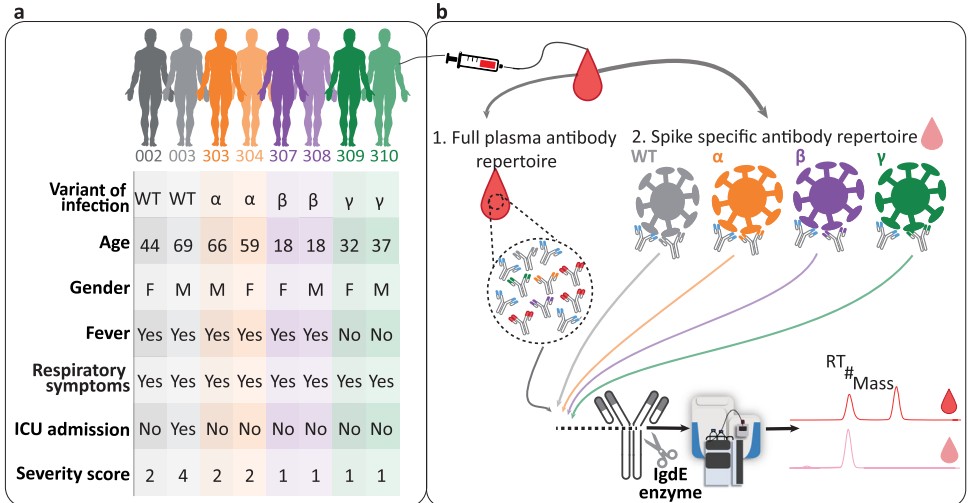

**Fig. 1 | Donor Characteristics and Monitoring of individual full IgG1 and S-protein antigen directed IgG1 profiles. a** Overview of the donors, who had experienced an infection by the named VOCs. VOCs are color-coded with WT (gray), Alpha (orange), Beta (purple) and Gamma (green). The table also lists age, gender, and disease state including Fever, respiratory symptoms and intensive care unit (ICU) admission following the World Health Care (WHO) severity score. **b** For each plasma sample taken we analyzed the full plasma IgG1 antibody repertoire as well as the antigen directed IgG1 clones to the four different VOCs S-proteins. The experimental approach involves the IgG capturing from full plasma as well as the S-protein specific immune-capturing. Fab fragments of the IgG1s were generated by enzymatic cleavage and subsequently subjected to intact-protein LC–MS analysis. Clonal repertoires could be profiled qualitatively, whereby each identified clone can be characterized by its unique accurate mass (in Dalton) and retention time (RT in minutes). By spiking in known quantities of two recombinant mAbs each plasma IgG1 clone could be quantified. The different S-protein specific Fab repertoires and the full plasma Fab repertoires were then compared, both intra- and inter-donors.

clones bind more restrictively to only one (or a few) variants, as we will discuss in further detail below.

## Cross-reactivity of IgG1 clones against VOC S-protein variants

Next, we focused on donor 003, who had developed most severe Covid-19, as indicated by the WHO severity score, and had been hospitalized after being infected with the WT-variant. For this donor, we were able to pull-down between 68 and 192 distinctive IgG1 clones with each of the S-protein variants tested, clearly indicating a quite broad polyclonal response (Supplementary Table 2). In Fig. 3a, b an overview of the ten most abundant S-protein directed IgG1 clones is given (with their identifier $^{RT}\#_{mass}$), depicting the quantity of each of the clones when affinity-enriched by each of the four S-protein variants. Clearly, the clone annotated as $^{19.2}64_{47,172.5}$ is the most abundant clone pulled down with the WT S-protein. This clone is also very abundant in the pull-down with the Alpha S-protein, but much less so following the pull-down with the Beta S-protein and is almost not captured when the Gamma S-protein is used. In contrast, other clones display wide and somewhat equal cross-reactivity against all four variants, such as $^{20.3}142_{47,792.9}$ and $^{18.3}248_{48,373.7}$. Yet another IgG1 clone is only pulled down with the WT and Gamma variant; $^{22.1}207_{48,689.7}$. In Fig. 3c a radar plot provides further insight in how each clone behaves in having distinctive affinity for the different S-protein variants. This data is compared with the radar plot obtained by summing all IgG1s having affinity for the S-proteins, as the latter is typically measured when total titers are assessed in donors. From this analysis it is directly clear that not every clone follows the same trend as observed for the total antigen directed IgG1 titer. Although we focused in this section on the responses of donor 003, alike observations were made for other donors, with generally each clone exhibiting its own pattern of reactivity against the four tested S-proteins.

Donor 303 is somewhat remarkable, in the sense that it harbors one clone $^{20.1}449_{48,655.2}$, with a more than 10 times higher affinity for the WT S-protein variant compared to Alpha, although the latter was the variant of infection. Again, there are also clones in this donor that show the same binding towards all S-protein variants. Another interesting trend for the clones found in this donor is, that the clones that

appear to bind to the Beta S-protein variant are barely showing binding to the Gamma S-protein variant and vice versa (Supplementary Fig. 6).

## Correlation of clonal repertoires with binding and neutralization

More standard approaches to evaluate antibody-dependent immune responses of individual donors use either Luminex bead-based binding assays or virus neutralization assays[9–12]. Thus, to corroborate our findings, we next performed such complementary neutralization and binding assays (Fig. 4). Neutralization was measured as $ID_{50}$ (plasma dilution that inhibits 50% of the infectivity) and the S-protein specific IgG binding was measured using a fluorescently labeled secondary antibody to detect S-protein binding IgG, resulting in a mean fluorescence intensity (MFI) (8, 14). To compare these results with the LC–MS based clonal profiling, we calculated the sum of the concentration of all detected IgG1 clones that were found to bind to the VOC specific S-protein variants in each donor (Fig. 4b). Of note, while the LC–MS based clonal profiling described above is IgG1 specific, the binding assay assess the full repertoire of IgGs (IgG1, IgG2, IgG3 and IgG4), and the neutralization assay the full plasma including the full repertoire of Igs (IgG, IgA, IgM) thus the results may be different, albeit that IgG1 is generally the most abundant subclass in plasma.

For donors 002, 003 and 304 the response to the VOC of infection dominates, as a priori expected, the radar plots in Fig. 4a. However, we also observe clear divergences. For example, donor 303 shows the highest neutralization and IgG binding for the variant of infection (Alpha), but the anti-S protein IgG1 concentration is much higher with the WT variant. For donor 307 the neutralization and specific IgG1 is much higher for the Gamma variant then for the Beta variant that infected this donor. Overall, there is quite some inconsistency between the three assays as seen in high total IgG binding vs low IgG1 specific binding (e.g. donor 304), or low neutralization vs high IgG1 binding (e.g. donor 308). However, also all this data discloses that Igs from a given donor, whereby infection as caused by a certain VOC, do cross-react quite well with viruses from other VOCs, thus widespread cross-reactivity is corroborated by all three assays.

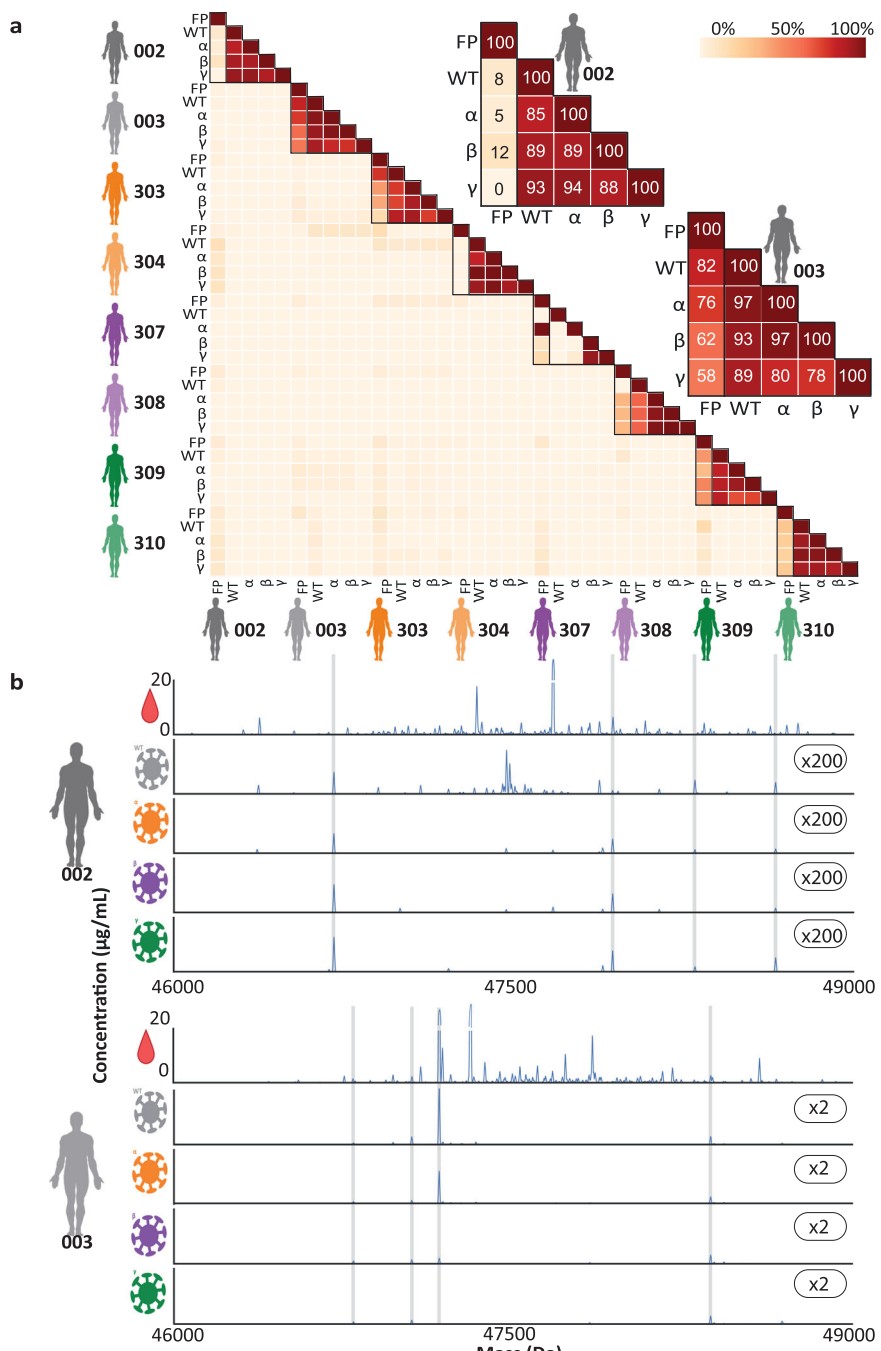

**Fig. 2 | S-protein specific IgG1 repertoires are polyclonal and unique per donor, whereas within a donor substantial cross-reactivity is observed when enriching with the four VOC S-protein trimers. a** Quantitative overlap of IgG1 repertoires illustrated by a heatmap, depicting the degree of overlap between all detected IgG1 repertoires, i.e. from all donors, extracted from either the full plasma (FP) or after pulldowns with each of the VOCs S-protein of all variants studied (WT, Alpha, Beta and Gamma). The quantitative overlap of Fab molecules, based on intensity per each unique identifier $^{RT}\#_{mass}$, was quantified and shown as a percentage as indicated by the color bar. The zoomed-in panels for donors 002 and 003 highlight the

substantial overlap for donor 003 between the S-protein specific Fab profiles, and the full plasma IgG1 profile Source data are provided as a Source Data file.
**b** Deconvoluted full Fab mass profiles as obtained for donor 002 and 003, from the S-protein specific Fab profiles (WT, Alpha, Beta and Gamma) and the full plasma Fab profile (top). Each peak represents a unique Fab at its detected mass and plasma concentration. The number on the right of each S-protein specific profile, indicated the y-axis multiplier compared to the y-axis used for the full plasma profiles. Supplementary Figs. 3–5 depict the full Fab mass profiles of all the other six donors.

## Discussion

The S-protein of SARS-CoV-2 is prone to mutations, stirring up questions regarding the protection of antibodies directed against one variant towards the other variants. Previous studies investigating the humoral immune responses of SARS-CoV-2 infected persons used either NGS[13–15] or ELISA[16–19] based methods. While NGS-approaches provide insights into specific clones at the DNA or RNA level, they do

not provide direct information about the abundances of these SARS-CoV-2 specific clones in circulation. The ELISA- based assays provide information on the produced Igs after SARS-CoV-2 infection, however, such assays lack the discriminatory power to provide insights at the level of individual unique antibody clones. Here we developed a direct MS-based approach enabling the analysis of the polyclonal response of individuals producing SARS-CoV-2 S-protein targeting IgG1 clones. Our

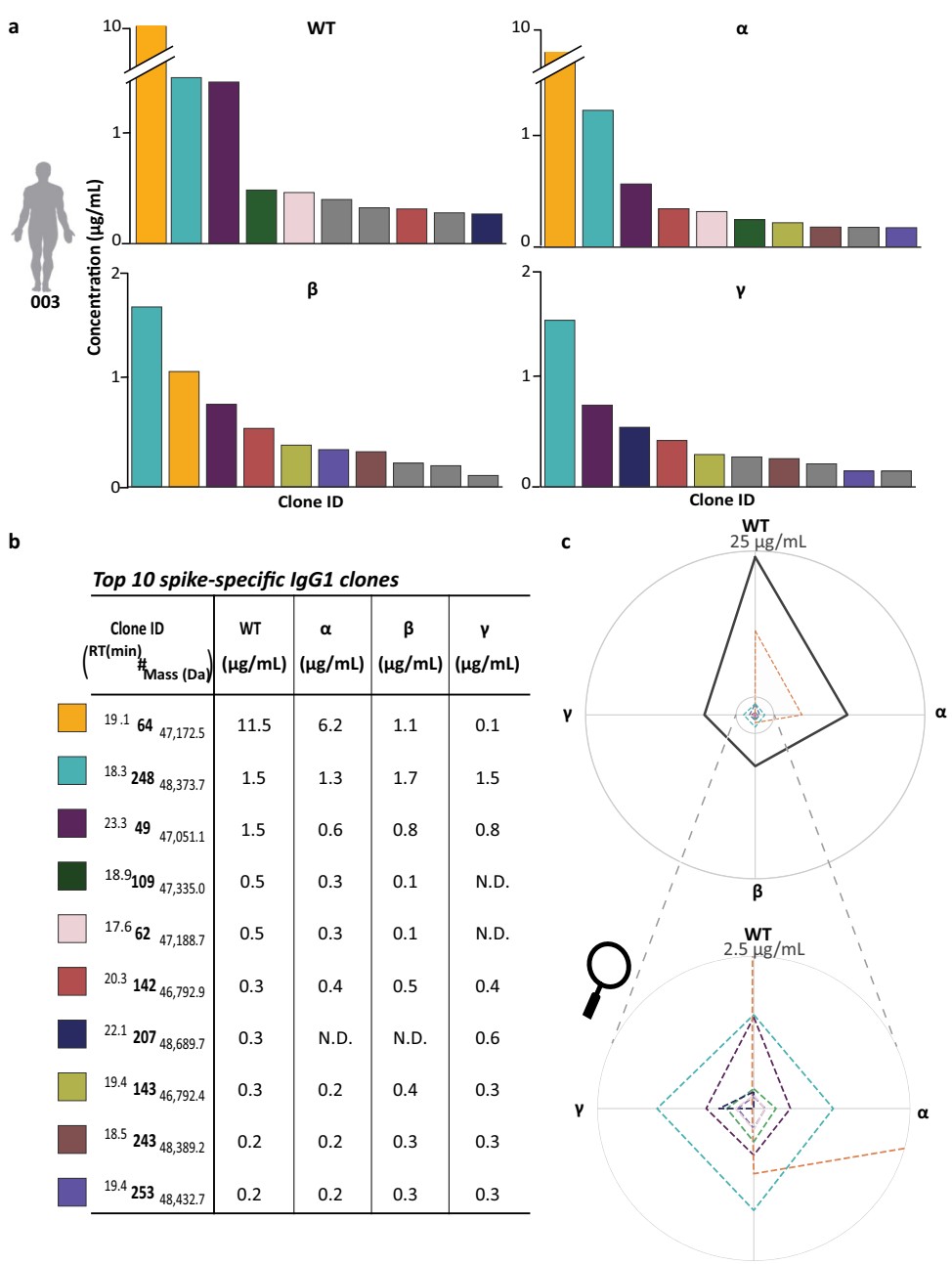

**Fig. 3 | Within a single donor, antigen directed IgG1 clones display distinctive cross-reactivity versus the VOCs S-protein variants. a** Quantitative comparison of the ten most abundant IgG1 Fab clones, affinity-enriched from the plasma of donor 003, with each of the four S-protein variants. Each bar represents one of the 10 most abundant clones with the height indicating the concentration in µg/ml, using the same colors for the clones corresponding to panel (**b**). Each clone that was not in the total top 10 but was in the top 10 for that specific VOC is colored grey. Source data are provided as a Source Data file. **b** Top 10 most abundant clones with their relative abundance after the pull-down with each of the four VOCs S-protein variants. The colored bars in **a** and the dotted lines in the radar plot in **c** are corresponding to the clones in the table B. **c** Radar plot with on each edge the data for one of the S-protein variants. These plots depict the difference in binding of specific clones against the VOCs S-protein variants. The thick solid black line representing the sum of all enriched IgG1 clones. Color coding is identical as used in **a** and **b**. Source data are provided as a Source Data file. The magnifying glass used in this figure was retrieved via Wikipedia.

data indicate that every donor shows a unique plasma IgG1 repertoire[7], but also a unique anti S-protein IgG1 repertoire, with each clone showing a distinct pattern of cross-reactivity versus different variants.

That each clone shows a unique binding pattern can be caused by the unique binding epitopes of the IgG clones in their interactions with the S-protein, leading to that some clones are more affected by specific mutations[20,21]. Ideally, we would know the exact epitopes of all anti-spike IgG1 clones we detect in our affinity pull-downs, but this would be very labor intensive and likely requires recombinant production of each of the clones. However, we can speculate about the epitopes of

each clone based on the (lack of) cross-reactivity and knowledge of specific mutations occurring in the different VOCs. Clearly, when a clone displays high cross-reactivity it is not affected by the mutations and may thus bind outside the regions affected by the mutations. Conversely, when a clone does not bind to one or two spike variants but does bind to the others this provide circumstantial evidence that this mutation may be part of the epitope.

In addition, factors such as avidity and (anti-) cooperativity are known to also influence binding between full-length IgG1 and the S-protein[22,23]. In any case, these binding differences between antibody

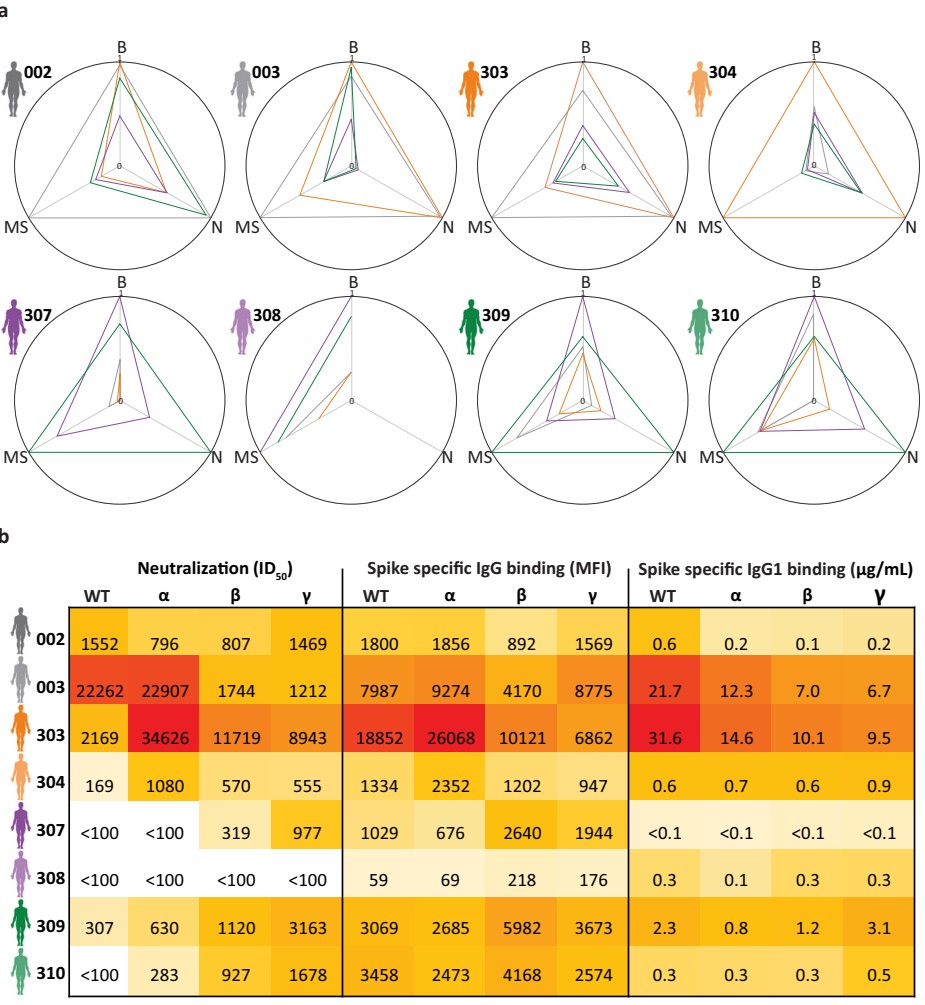

**Fig. 4 | Properties of antigen specific IgG(1) repertoires and correlation between different binding and neutralization assays. a** each radar plot represents data for a single donor, with axis representing data of the neutralization assay (N), S-protein specific total IgG binding assay (B) and S-protein specific IgG1 binding by LC−MS (MS). Per axis and per donor, the values are normalized based on the highest value observed for that axis for that specific donor, giving a value 1 with the other values displaying the proportion of this highest base value, i.e. in between 0 and 1. Each line represents data against a particular S-protein (shown with the color of the line) for that specific donor. **b** Values behind the radar plots in A. $ID_{50}$ determined in the neutralization assays (left), and amount of S-protein specific IgG binding observed by the fluorescence assay (middle) and the LC−MS based clonal profiling (right). In the latter assay S-protein specific binding is calculated by summing up the concentrations of the S-protein specific clones for that specific donor. The colors represent the high (in red) to low (light yellow) values determined per individual assay.

clones are an important feature to consider when selecting antibodies for further development into biotherapeutics. Here we show that not always the most abundant antibody exhibits the best cross-reactivity against all other S-protein variants.

Our data show that IgG binding does not always correlate with neutralization what may suggest alternative functions for these S-protein binding Igs. Moreover, there is evidence from the literature that antibodies against SARS-CoV-2 with low IgG fucosylation result in increased macrophage activation, thereby introducing the antibody dependent cellular cytotoxicity (ADCC) and phagocytose and with that highlighting the broad functionality of antibodies also during and after SARS-CoV-2 infection[24,25]. Besides, whether a specific antibody against SARS-CoV-2 will neutralize or not is likely also epitope dependent[20,26]. The difference we observe for some donors between the MS-based IgG1 binding method and the neutralization and Luminex binding assay could possibly originate from the fact that with the MS-based method we are assessing solely IgG1 binding while neutralization and the luminex binding assays are performed on total plasma and total IgG level, respectively. These differences may suggest that the spike

specific immune response for these donors is dominated by IgG subclasses other than IgG1. Next to IgG1, IgG3 has been reported to be broadly involved during SARS-CoV-2 infection[27–29]. These differences observed between donors, again, highlight the unique donor-specific humoral immune responses towards SARS-CoV-2. For future studies it would be interesting to not only focus on the IgG1s, but also look at the clonal profiles of other IgG subclasses e.g. IgG2, IgG3 and IgG4. That donor 307 and 308 show the lowest amount of spike-binding antibodies can potentially be explained by their relatively young age (both 18 years old) combined with their lowest severity score.

Although here we assessed the polyclonal IgG1 responses in donors suffering a SARS-CoV-2 infection, our direct LC−MS method may equally well be exploited to monitor responses induced by infections with other pathogens like viruses and bacteria, and even certain diseases such as cancer and rheumatoid arthritis.

Moreover, although this study was restricted to the VOCs known at the time of conducting the experiments, it can be extended to other and new VOCs. Our data are in agreement with the finding that humans may (have) acquire(d) pre-existing cross-reactive humoral immunity,

mostly against the S2 subunit of the S-protein, as this part is mostly conserved between the different S-protein variants of SARS-CoV-2 and among the sequences of related group B coronaviruses[20,30–32]. It is also the S1 subunit that is shown to be mostly mutated in the Delta and Omicron VOCs, suggesting that the pre-existing cross-reactivity against the S2 subunit is not that much affected. This, together with the unique S-protein specific response we observed for each donor, could explain why some individuals' repertoires better protect against re-infection with another VOC than others.

In conclusion, by assessing IgG1 repertoires following infection we observe a widespread polyclonal IgG1 response in SARS-CoV-2 infected people that are unique for each person. Each clone exhibited a distinct pattern of cross-reactivity *versus* SARS-CoV-2 S-protein variants. Furthermore, the clonal repertoire analysis did not per se correlate with in vitro neutralization assays, highlighting the need for a variety of assays to judge the full scale of antiviral fitness of a patient sample. The knowledge that every clone shows a different SARS-CoV-2 bindings pattern, is important to consider when developing new biotherapeutics or novel vaccination strategies. And while here shown for SARS-CoV-2, the here described approach can be further utilized to have a detailed look into the antigen-specific antibody response in a wide variety of diseases.

## Methods

### Donor characteristics

The study was conducted at the Amsterdam University Medical Centers, location AMC, in the Netherlands and approved by the local ethical committee of the AMC (NL 73281.018.20). All individuals included in this study gave written informed consent before participating. The eight donors included were part of the larger cross-sectional COSCA cohort (NL 73281.018.20) as described previously[9]. The plasma of these infected adults were collected 3–6 weeks after symptoms onset. All participants had at least one nasopharyngeal or oropharyngeal swab positive for SARS-CoV-2, for half of them (COSCA-303, 308, 309 and 310) the variant of infection was sequence confirmed. For the other donors the variant of infection was assumed by a proven variant of infection by their household member (COSCA-304 and 307) or because no VOC had emerged yet at the time of sampling (COSCA-002 and 003). Participants were included from the start of the COVID-19 pandemic in the Netherlands in March 2020 until the end of February 2021. None of the included donors received any vaccination against SARS-CoV-2 prior to the study.

### Production and purification of VOC spike protein trimer variants

The S-protein antigen constructs representing the different VOCs contained the following mutations compared to the WT variant (Wuhan Hu-1; GenBank: MN908947.3): deletion (Δ) of H69, V70 and Y144, N501Y, A570D, D614G, P681H, T716I, S982A, and D1118H in Alpha (B.1.1.7); L18F, D80A, D215G, L242H, R246I, K417N, E484K, N501Y, D614G, and A701V in Beta (B.1.351); and L18F, T20N, P26S, D138Y, R190S, K417T, E484K, N501Y, D614G, H655Y, and T1027I in Gamma(P.1). The genes were ordered as gBlock gene fragments (Integrated DNA Technologies) and cloned Pst I/Not I in a pPPI4 expression vector containing a hexahistidine (his) tag with Gibson Assembly (Thermo Fisher Scientific). All S constructs were verified by Sanger sequencing and the protein was subsequently produced in human embryonic kidney (HEK) 293 F cells (Thermo Fisher Scientific) and purified as previously described[4,9].

### SARS-CoV-2 S-protein trimer specific antibody enrichment

The full-length trimeric S-protein variants of the different VOCs were covalently bound to Pierce NHS-Activated Agarose Spin Columns. Four Pierce NHS-activated agarose spin columns, each loaded with a different VOC S-protein variant, were assembled according to

manufacturer's instructions, and placed in 2 mL Eppendorf Tubes. Therefore, each of the spin columns was incubated with 0.5–1.0 mg of one of the variants (either WT, Alpha, Beta or Gamma). The S-proteins together with the beads were then incubated for 2 h using an end-over-end rotator at room temperature. After incubation with the S-protein, the flowthrough was collected, and the agarose spin columns were washed two times with 400 μl PBS. After the washing steps, the agarose spin columns were incubated with 400 μl Tris (1 M, pH 8) for 30 min as a quenching buffer, using an end-over-end rotator at room temperature. After incubation with Tris the columns were washed three times with 400 μl PBS and subsequently stored at 4 degrees in 300 μl PBS. 60 μl S-protein-bead slurry was incubated with 200 μL of each plasma sample separately (marked as COSCA002, 003. 303, 304, 307, 308, 309 and 310) together with 300 μl PBS in Pierce spin columns (ThermoFisher Scientific). The different plasma samples were each incubated with the different S-protein-bead variants for 2 h (e.g., eight different plasma samples with the four different VOCs SARS-CoV-2 S-protein variants) on an end-over-end rotator at room temperature. After incubation with the SARS-CoV-2 S-proteins the flowthrough was collected, further referred to as the unbound fraction. Subsequently, the spin columns were washed two times with 600 μl PBS and two times with 600 μl Milli-Q water. Hereafter, the SARS-CoV-2 spike specific antibodies could be eluted from the S-proteins-beads. To this end, we added 100 μL Glycine-HCl (pH 2.7), incubated shaking for 10 min, and collected the so-called bound fraction by centrifugation for 1 min at 500 ×g. This was repeated two more times. The bound fraction was captured in a 1.5 mL Eppendorf already containing 60 μl Tris (1 M, pH 8) to be able to directly neutralize the eluted fractions. As an validation test we used the same approach using 200 ul Plasma of Donor 003 as input, with the only difference that we used 'bare' NHS agarose beads (beads without any of the spike variants bound), enabling the assessment of non-specific binding to the beads (Supplementary Fig. 1). Additionally we tested the specificity of the spike-bead using the WT S-protein beads with plasma of Donor 003 in which we spiked an earlier described anti-Spike mAb (10 μg/ml COVA 2–15 IgG1[9]) and as a control 10 μg/ml IgG1 Bevacizumab (anti-VEGF). This shows that the anti-spike mAb bound to these beads and could be retrieved (~85%), whereas no Bevacizumab could be retrieved (Supplementary Fig. 2).

### IgG purification, and subsequent Fab generation

For the IgG purification and the generation of Fabs, a similar protocol was used as described earlier albeit with some adaptations[7]. In short, 20 μl FcXL affinity matrix slurry was directly added to Pierce spin columns (ThermoFisher Scientific), followed by three washing steps with 150 μl Phosphate buffer (PB, 150 mM, pH7), in which for each washing step the liquid was removed by centrifugation for 1 min at 500 ×g at room temperature. After washing, the 2 mL tube was replaced by a 1.5 mL tube. The affinity matrix was resuspended in 150 μl PB for the unbound and full plasma fraction and in 40 μl PB with a 1% blocking buffer background (Bio-Rad, The Netherlands) for the bound fractions. Subsequently 50 μl of the unbound fraction, 20 μl of full plasma or the whole bound fraction (360 μl) were added to the corresponding affinity matrix. Furthermore, 1 μl of a solution containing two monoclonal antibodies (mAbs) (i.e., trastuzumab and alemtuzumab) at 200 μg/mL each, was added as internal standard for quantification to the unbound and plasma fraction. Alternatively, 1 μl of a solution containing these mAbs at 50 μg/mL was added to the bound fraction. The samples were then incubated under shaking conditions for one hour at room temperature. After incubation, the flowthrough was collected and the affinity matrix with bound IgGs was washed four times with 200 μl PB. Finally, 50 μl PB containing 50 U of immunoglobulin degrading enzyme (IgdE; branded FabALACTICA, Genovis AB, Lund, Sweden) was added, which selectively cleaves only IgG1s in their hinge region, before incubation on a

thermal shaker at 37 °C for at least 16 h. After overnight incubation with IgdE, the flowthrough containing the Fab fragments generated from the bound IgG1s was collected by centrifugation for 1 min at 500 ×g. Using this approach, we generated IgG1 Fab samples from full plasma, and from the bound and unbound fraction for all included donors, following affinity purification with any of the four VOCs S-protein variants.

## LC–MS profiling

For the mass analysis of the intact released Fabs an LC–MS and data processing approach was used as described by Bondt et al[7]. In short, the collected intact Fab proteins were separated by using a Thermo Scientific Vanquish Flex UHPLC instrument, equipped with a 1 × 150 mm MAbPac Reversed Phase HPLC Column. Both the column preheater and the analytical column chamber were heated to 80 °C during chromatographic separation. The LC was directly coupled to an Orbitrap Exploris 480 mass spectrometer with BioPharma option (Thermo Fisher Scientific, Bremen, Germany). The Fab samples were separated over a 62 min gradient at a 150 μl/min flow rate. Gradient elution was achieved using two mobile phases, A (0.1% HCOOH in Milli-Q water) and B (0.1% HCOOH in CH3CN) at a starting mixture of 90% A and 10% B, ramping up from 10% to 25% over 1 min, from 25% to 40% over 54 min, and from 40% to 95% over 1 min. MS data were collected with the instrument operating in intact protein and low-pressure mode. Spray voltage was set at 3.5 kV from minute 2 to minute 50 to prevent salts in the sample from entering the MS, ion transfer tube temperature was set at 350 °C, vaporizer temperature at 100 °C, sheath gas flow at 15 arb.units, auxiliary gas flow at 5 arb.units, and source-induced dissociation (SID) was set at 15 V. Spectra were recorded with a resolution setting of 7500 (@m/z 200) in MS1 allowing improved detection of charge distributions of large proteins (>30 kDa)[33]. Scans were acquired in the range of 500 – 4000 $m/z$ using an automated gain control (AGC) target of 300% and a maximum injection time set to 50 ms. For each scan 5 μscans were recorded.

## Data analysis

The retention times and masses of each of the Fab molecules were retrieved from the generated RAW files using BioPharmaFinder 3.2 (Thermo Scientific). Deconvolution was performed using the ReSpect algorithm between 5 and 57 min using 0.1 min sliding windows with 25% offset and a merge tolerance of 30 ppm, and noise rejection set at 95%. The output range was set at 10,000–100,000 Da with a target mass of 48,000 Da and mass tolerance of 30 ppm. Charge states between 10 and 60 were included, and the Intact Protein Peak model was selected. Further data analysis was performed using in-house scripts using Python 3.8.3 (with libraries: Pandas 1.0.5, Numpy 1.18.5, Scipy 1.5.0, matplotlib 3.2.2 and seaborn 0.11.0). Masses of the Bio-PharmaFinder identifications (components) were recalculated using an intensity weighted mean, considering only the most intense peaks comprising 90% of the total intensity. Furthermore, using the data of the two spiked-in recombinant mAbs (i.e., trastusumab and alemtuzumab) the intensity could be normalized, and each Fab clonal signal quantified to a concentration in μg/ml.

Components between 45,000 and 53,000 kDa with the most intense charge state above m/z 1000 and BPF score >=40 were considered likely Fab fragments of IgG1 clones. Clones within 1.4 Da mass and 0.8 min retention-time window were considered identical. These windows are defined as three times the standard deviation of the signals obtained for the mAb standards that were spiked in, which was 1.4 Da for the mass and 0.8 min for the retention time.

## Binding and neutralization assay

**Protein coupling to Luminex beads**. To measure the binding of IgG to the S-proteins of different VOCs, we covalently coupled pre-fusion stabilized S-proteins to Luminex Magplex beads using a two-step carbodiimide reaction as previously described[10]. In short, Luminex Magplex beads (Luminex) were washed with 100 mM monobasic sodium phosphate pH 6.2 and activated by addition of Sulfo-N-Hydroxysulfosuccinimide (Thermo Fisher Scientific) and 1-Ethyl-3-(3-dimethylaminopropyl) carbodiimide (Thermo Fisher Scientific) and incubated for 30 min on a rotator at room temperature. After washing the activated beads three times with 50 mM MES pH 5.0, the S-proteins were added in ratio of 75 μg protein to 12.5 million beads and incubated for three hours on a rotator at room temperature. To block the beads for aspecific binding, we incubated the beads for 30 min with PBS containing 2% BSA, 3% fetal calf serum and 0.02% Tween-20 at pH 7.0. Finally, the beads were washed and stored at 4 °C in PBS containing 0.05% sodium azide[31,34].

**Luminex assays**. Optimization experiments declared the optimal concentration of the plasma for studying the humoral response following SARS-CoV-2 infection to be 10.000-fold dilution. As previously described[10,31], 50 μL of a bead mixture containing all different S-proteins in a concentration of 20 beads per μL were added to 50 μL of diluted plasma and incubated overnight on a rotator at 4 °C. The next day, plates were washed with TBS containing 0.05% Tween-20 (TBST) and resuspended in 50 μL of Goat-anti-human IgG-PE (Southern Biotech) with a concentration of 1.3 μg/mL. After 2 h of incubation on a rotator at room temperature, the beads were washed with TBST and resuspended in 70 μL Magpix drive fluid (Luminex). Read-out of the plates was performed on a Magpix (Luminex). The binding of antibodies is expressed as the Median Fluorescence Intensity (MFI) of approximately 50 to 100 beads per well. MFI values are corrected for background signals by subtracting the MFI of wells containing only buffer and beads.

**Pseudo-virus construction**. The WT, Alpha, Beta and Gamma pseudovirus S constructs were ordered as gBlock gene fragments (Integrated DNA Technologies) and cloned using SacI and ApaI in the pCR3 SARS-CoV-2-SΔ19 expression plasmid[35] using Gibson Assembly (ThermoFisher). All constructs were verified by Sanger sequencing. Pseudo-viruses were produced by co-transfecting the SARS-CoV-2-S expression plasmid with the pHIV-1NL43 ΔEnv-NanoLuc reporter virus plasmid in HEK293T cells (ATCC, CRL-11268), as previously described[35]. Cell supernatant containing the pseudo-virus was harvested 48 h post transfection and stored at −80 °C until further use.

**Pseudo-virus neutralization assays**. Neutralization activity was tested using a pseudo-virus neutralization assay, as previously described[4]. Shortly, HEK293T/ACE2 cells, kindly provided by Dr. Paul Bieniasz[35], were seeded at a density of 20,000 cells/well in a 96-well plate coated with 50 μg/mL poly-L-lysine one day prior to the start of the neutralization assay. NAbs (1–50 μg/mL) or heat-inactivated plasma samples (1:100 dilution) were serially diluted in cell culture medium (DMEM (Gibco), supplemented with 10% FBS, penicillin (100 U/mL), streptomycin (100 μg/mL) and GlutaMax (Gibco)), mixed in a 1:1 ratio with pseudo-virus and incubated for 1 h at 37 °C. Subsequently, these mixtures were added to the cells in a 1:1 ratio and incubated for 48 h at 37 °C, followed by a PBS wash and lysis buffer to measure the luciferase activity in cell lysates using the Nano-Glo Luciferase Assay System (Promega) and GloMax system (Turner BioSystems). Relative luminescence units (RLU) were normalized to the positive control wells where cells were infected with pseudo-virus in the absence of NAbs or plasma. The inhibitory concentration (IC$_{50}$) and neutralization titers (ID$_{50}$) were determined as the NAb concentration and plasmadilution at which infectivity was inhibited by 50%, respectively, using a non-linear regression curve fit (GraphPad Prism software version 8.3). Samples with ID$_{50}$ titers <100 were defined as having undetectable neutralization.

**Reporting summary**

Further information on research design is available in the Nature Research Reporting Summary linked to this article.

## Data availability

The mass spectrometry data used in this study has been deposited to the MassIVE repository (https://massive.ucsd.edu/ProteoSAFe/static/massive.jsp) under accession code MSV000089833 (https://massive.ucsd.edu/ProteoSAFe/dataset.jsp?task=9bcf87169a7e4a849e0d254fe80b3828). The source data underlying Figs. 2a and 3a, 3c and Supplementary Figures 6A and 6C are provided as Source data. Source data are provided with this paper.

## Code availability

The data analysis code used in this study has been deposited to the MassIVE repository (https://massive.ucsd.edu/ProteoSAFe/static/massive.jsp) under accession code MSV000089833.

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

## Acknowledgements

This research received funding through the Dutch Research Council (NWO) funding the Netherlands Proteomics Centre through the X-omics Road Map program (project 184.034.019) and Gravitation Subgrant 00022 from the Institute for Chemical Immunology (D.M.H.v.R. and A.B.). A.J.R.H. acknowledges support from the Netherlands Organization for Scientific Research (NWO) through the Spinoza Award SPI.2017.028 to A.J.R.H., R.W.S. acknowledges support from the Netherlands Organization for Scientific Research (NWO) through a Vici grant 91818627.

The authors also acknowledge support from the Bill & Melinda Gates Foundation through grant INV-024617 (M.J.v.G.). M.J.v.G. is a recipient of an Amsterdam UMC AMC Fellowship.

## Author contributions

D.M.H.v.R, A.B., M.J.v.G., and A.J.R.H. came up with the concept and designed the research; K.v.d.S., D.E, C.R., R.W.S., and G.J.d.B set up the COSCA study and recruited the study participants; D.M.H.v.R., A.B, K.v.d.S., T.G.C. and M.P. performed experiments; D.M.H.v.R., A.B., M.H., K.v.d.S. and A.J.R.H analyzed the data; and D.M.H.v.R., A.B., and A.J.R.H. wrote the paper, which was edited by all co-authors.

## Competing interests

The authors declare no competing interests.
