## [Peer Review File · Nature Communications]

Discriminating cross-reactivity in polyclonal IgG1 responses against SARS-CoV-2 variants of concernREVIEWER COMMENTS

Reviewer #1 (Remarks to the Author):

Review

IgG1 responses following SARS-CoV-2 infection are 2 polyclonal and highly personalized, whereby each donor 3 and each clone displays a distinct pattern of cross4 reactivity against SARS-CoV-2 variants

By Danique M.H. van Rijswijck et al

The authors present a unique IgG1 antibody characterization that builds on the groups previous work of characterizing the polyclonal IgG 1 background of normal donors. The method was applied to patients who were previously infected with SAR-Covid. As far as I know, this work is unique in its approach to Covid antibody response. Given the small cohort of patients and limitation of only characterizing IgG1, general conclusions about the immune response cannot be made. However, I found great merit in the novel mass spectroscopic method used to characterize the IgG1 repertoires. I agree with the authors that this type of measurements will eventually become common place in research focused on antibody responses and I would like to see it published. I found the figures to be clear and helpful and the writing is clear. I do have some follow up questions/data that I would like to explore prior to publication.

1. In my experience, the pre-analytical sample preparation is key to getting representative clonal distributions.
 - a. How did the authors assure that the S-protein beads did not allow for non-specific binding of immunoglobulins? Were negative control beads tested? Was any work done on optimizing the bead washing steps to assure removal/retention of low affinity IgG clones? Any estimation of the binding capacity of the beads performed and if so, were the estimated total anti-S-protein Ig concentrations near the binding capacity. Did the authors attempt any competitive
2. Line 457 pg 18 Clones within a 1.4 Da mass and 0.8 min retention-time window were considered identical. How was this determined?
3. Were the S-protein constructs bound to an ELISA plate and tested for relative testing?
4. I my personal but limited experience with earlier orbitrap applications to polyclonal Igs, I have seen instances in which clones were missed (presumably due to sampling times) that were apparent on Q-TOF measurements. I would like to hear the authors comments on this.
5. Was the Fc portion examined at all? The glycosylation patterns on the Fc may also be interesting?
6. In figure 2, I understood the discussion of the overlap of the patients total IgG1 with the anti-s-Protein, but I did not see page 6 "For some donors the S-protein directed IgG1 repertoires are within that donor quite alike (e.g. donors 003, 303, 304 and 310), no matter which S-protein had been used as affinity handle. I understand this as the Fab masses and retention times of the clones are the same between donor 003,303, 304 and 310. Please clarify for me.

Minor

1. I am assuming none of the study participants had received vaccination prior to the study. I think this is an important negative to mention.

David Murray

Reviewer #2 (Remarks to the Author):

This study used a mass-spectrometry (MS)-based method to investigate IgG1 repertoire of 8 COVID-19 patients. They found unique profiles for patients. This mass-spectrometry based approach is complementary to more traditional ways in identifying the antibody repertoire.

Major comments:

- How reproducible is the MS-based method? For novel assays, it would be important to perform multiple runs on the same samples to assess for reproducibility.

- Has the authors conducted any evaluation of the MS-based method for spike proteins? For example, have they used monoclonal antibodies against the spike protein as positive controls and monoclonal antibodies targeting another virus as negative controls (eg. a monoclonal antibody against influenza virus)?
 - For each of the clones identified, is it possible to identify the actual binding epitope on the spike protein? This would be especially important if the clone can bind to different variants and may be selected for further development into therapeutic monoclonal antibodies.
 - The authors claimed that their study can “aid in selecting antibodies that may be developed into biotherapeutics. However, since these antibodies are identified by mass spectrometry, the original B cells that produce these antibodies would not be known. Without the original B cells, how can they produce monoclonal antibodies which are exactly the same as the ones identified by the mass spectrometry? Please explain.
 - In line 177-180, the authors suggest that for donor002, the majority of the abundant clones are not directed to S protein (line 177-180). However, in Figure 4B, donor002 has relatively high neutralizing antibody titer against all variants and binding antibody titer against the S protein. Please explain the apparent discrepancy between the MS-based method and the neutralization/binding antibody assays.
- Minor comments:
- Patient details: Were any of these patients admitted to ICU or require oxygen supplementation? Any acute complications? Was steroid or other immunomodulators given during hospitalization?
 - Figure 1 and Supplementary Table 1. For all abbreviations, provide the full name. For example, what does IC stand for?
 - For supplementary Table 1, it would be useful for readers to outline the definition of the WHO scores.

Reviewer #3 (Remarks to the Author):

In the manuscript entitled “IgG1 responses following SARS-CoV-2 infection are polyclonal and highly personalized, whereby each donor and each clone displays a distinct pattern of cross-reactivity against SARS-CoV-2 variants”, the authors used MS based analytical method to discover the molecular diversity and quantity of IgG1 from 8 donors after their infection of SARS-CoV-2 and variants. The topic is important to be investigated, and some comments are listed in the following:

1. Cross-reactivity of IgG1 from donors infected by one specific strain to other variants was well discussed and investigated in this study; however, only half of the donors got sequence-confirmed results for the infected strains. This should be indicated when performing the results, for example, on page 9 and line 258~267.
2. As different spike proteins were used to enrich antibodies from donor plasma, specific mutations for each VOC on the spike protein are suggested to be mentioned in the introduction or discussion.
3. As mentioned by the authors, other IgG subclasses might also play an important role in SARS-CoV-2 infection, such as IgG3. Therefore, the isotypes or subclasses information for antigen-enriched immunoglobulins are suggested to be profiled as supporting information for this study.
4. Line 253~255 and figure 4: if the three methods target different immunoglobulin populations, can they be discussed or compared together? I hope the experiment suggested in the third point will help to clarify this issue.
5. Among the eight donors, participants 307 and 308 showed the least functional antibodies (figure 4B). Is there any comment?
6. Many studies have characterized B cell receptor and humoral immune system to the COVID-19. They should be included in the discussion and compared to the current study too. In addition, why the authors selected IgG1 to investigate the diversity should be introduced in the work.

REVIEWER COMMENTS with our responses in green

Reviewer #1 (Remarks to the Author):

Review

IgG1 responses following SARS-CoV-2 infection are 2 polyclonal and highly personalized, whereby each donor 3 and each clone displays a distinct pattern of cross4 reactivity against SARS-CoV-2 variants
By Danique M.H. van Rijswijck et al

The authors present a unique IgG1 antibody characterization that builds on the groups previous work of characterizing the polyclonal IgG 1 background of normal donors. The method was applied to patients who were previously infected with SAR-Covid. As far as I know, this work is unique in its approach to Covid antibody response. Given the small cohort of patients and limitation of only characterizing IgG1, general conclusions about the immune response cannot be made. However, I found great merit in the novel mass spectroscopic method used to characterize the IgG1 repertoires. I agree with the authors that this type of measurements will eventually become common place in research focused on antibody responses and I would like to see it published. I found the figures to be clear and helpful and the writing is clear. I do have some follow up questions/data that I would like to explore prior to publication.

We very much appreciate the appreciative comments by Dr David Murray and like to thank him as a real expert for taking the time to review our work.

1. In my experience, the pre-analytical sample preparation is key to getting representative clonal distributions.

a. How did the authors assure that the S-protein beads did not allow for non-specific binding of immunoglobulins? Were negative control beads tested? Was any work done on optimizing the bead washing steps to assure removal/retention of low affinity IgG clones? Any estimation of the binding capacity of the beads performed and if so, were the estimated total anti-S-protein Ig concentrations near the binding capacity. Did the authors attempt any competitive

These comments are all very valid, and indeed, we did perform these validation tests prior to the work described, but maybe they were not fully described in the manuscript. To address these comments, we provide in the revised manuscript a Supplementary Figure 1, depicting data we obtained from the negative control test with the 'bare' NHS agarose beads. For this control, we used the approach depicted in Figure 1 using 200 µl Plasma of Donor 003 as input, with the only difference that we used NHS-activated agarose beads without any of the spike variants bound, enabling the assessment of non-specific binding to the beads. We concluded that there was no non-specific binding to the agarose beads. See line 96-101, 334-337.

In our analysis we aimed for a robust protocol to allow qualitative comparison of valuable antibodies targeting the various S-protein variants, which are in the case of IgG often of high affinity. Therefore, we approached this as any affinity enrichment assay, and did not further optimize to prevent the loss of low affinity clones.

By empirical assessment, we determined that the binding capacity of the spike-beads was sufficient for the anti-spike IgG1 present in most of the samples, namely those of donors that were not hospitalized. For the hospitalized donor 003, whose plasma contained high levels of anti-spike IgG1, we observed that

the highest clone (clone ^{19.1}64_{47,172.5} in Figure 3) was also observed in the ‘unbound’ fraction, indicating insufficient binding capacity. In contrast, we did not find any of the clones of the ‘bound’ fraction back in the ‘unbound’ fraction when evaluation for instance donor 002, who had lower levels of anti-spike IgG1 in their plasma. To obtain as much qualitative information as possible from the available material we settled with 200 µL plasma for all donors. For the current manuscript we felt that the qualitative comparison (e.g. which clones) was more important than quantitative (how much of the specific clone).

2. Line 457 pg 18 Clones within a 1.4 Da mass and 0.8 min retention-time window were considered identical. How was this determined?

Indeed, this choice appears to be a bit arbitrary. Our rationale was to define these windows as three times the standard deviation of the signals we obtained for the mAb standards that were spiked in, which was 1.4 Da for the mass and 0.8 min for the retention time. We now clarified our rationale in the methods section of the revised manuscript (see line 404-407).

3. Were the S-protein constructs bound to an ELISA plate and tested for relative testing?

Indeed, this was tested but instead of an ELISA, we performed Luminex-based binding assays. For this, the S-protein constructs were covalently coupled to Luminex Magplex beads (Luminex) as described in lines 410-421. Using this approach IgG binding of the different SARS-CoV-2 infected donors was assessed (see results from line 182 onwards).

4. I my personal but limited experience with earlier orbitrap applications to polyclonal Igs, I have seen instances in which clones were missed (presumably due to sampling times) that were apparent on Q-ToF measurements. I would like to hear the authors comments on this.

This is a detailed mass spectrometry questions and we think we know what the reviewer means. On an Orbitrap-based mass analyzers the users can choose to set certain parameters (e.g. low or high resolution, accumulation time, ion loading etc.) that influence detection. Using high-resolution settings, the mass resolution may become better, but this may come at the expenses of not detecting lower abundant peaks/clones. Here we optimized the settings especially for detection Fab molecules, and with these settings, we detect many more clones by using the Orbitrap compared to when using the Waters Q-ToF, although in our lab the latter is quite an old one. We feel this may answer the question, but rather chose not to comment on this in the manuscript, as it is beyond the scope.

5. Was the Fc portion examined at all? The glycosylation patterns on the Fc may also be interesting?

We agree with the reviewer that the glycosylation patterns of the Fc region of these donors are also interesting to investigate, but that would acquire a different approach. As we were interested in the polyclonal response, we focused solely on the Fabs. In addition, as we use FcXL beads for capturing, we get a pool of the Fc regions of all IgG subclasses that would make it rather difficult to draw conclusions about the Fcs belonging to the clones we see back in our IgG1 clonal profile.

6. In figure 2, I understood the discussion of the overlap of the patients total IgG1 with the anti-s-Protein, but I did not see page 6 “For some donors the S-protein directed IgG1 repertoires are within that donor quite alike (e.g. donors 003, 303, 304 and 310), no matter which S-protein had been used as

affinity handle. I understand this as the Fab masses and retention times of the clones are the same between donor 003,303, 304 and 310. Please clarify for me.

We feel indeed this sentence could be more clarified and have changed it into: “For some donors (e.g. donors 003, 303, 304 and 310) the S-protein directed IgG1 repertoires are within a single donor quite alike, no matter which S-protein had been used as affinity handle.” (Lines 139-141).

Minor

1. I am assuming none of the study participants had received vaccination prior to the study. I think this is an important negative to mention.

Valid point, indeed none of the donors had received a SARS-CoV-2 vaccination prior to the study, we mentioned this in the revised manuscript (lines 293-294):

“None of the included donors received any vaccination against SARS-CoV-2 prior to the study.”

David Murray

Reviewer #2 (Remarks to the Author):

This study used a mass-spectrometry (MS)-based method to investigate IgG1 repertoire of 8 COVID-19 patients. They found unique profiles for patients. This mass-spectrometry based approach is complementary to more traditional ways in identifying the antibody repertoire.

Major comments:

- How reproducible is the MS-based method? For novel assays, it would be important to perform multiple runs on the same samples to assess for reproducibility.

We agree with the reviewer that it is very important that the here used MS-based method is reproducible. In our earlier work (Bondt et al. Cell Systems 2021), we described, when developing the method, how we tested the reproducibility in terms of technical replicates and sample replicates. These validation experiments revealed very clearly that, by using the MS-based method used here, the IgG1 Fab clonal repertoires could be reproducibly determined. Therefore, we do feel that we should not reproduce these tests in the current manuscript. But to address this point we added to the revised text (lines 101-103): *“Previously we already showed that the here presented MS-based method results in highly reproducible data on IgG1 repertoires, both in biological and analytical replicates (Bondt et al. Cell Systems 2021).”*

- Has the authors conducted any evaluation of the MS-based method for spike proteins? For example, have they used monoclonal antibodies against the spike protein as positive controls and monoclonal antibodies targeting another virus as negative controls (eg. a monoclonal antibody against influenza virus)?

In response to the reviewer, we have included now the following experiment in the revised manuscript. Using the approach depicted in Figure 1B, we used the WT S-protein beads with plasma of Donor 003 in which as a positive control we spiked an earlier described anti-spike mAb (10 µg/ml COVA 2-15 IgG1, see Brouwer et al. Science 2022), and as a negative control we included 10 µg/ml IgG1 Bevacizumab (anti-VEGF). We provide a new Supplementary Figure 2 revealing that the anti-spike mAb bound to these beads and could be retrieved (~85%), whereas no Bevacizumab could be retrieved. See text lines 96-101, 337-342.

- For each of the clones identified, is it possible to identify the actual binding epitope on the spike protein? This would be especially important if the clone can bind to different variants and may be selected for further development into therapeutic monoclonal antibodies.

In theory this should be possible and interesting, albeit this would be a demanding task and beyond the scope of our work. For this, we would need to make all clones recombinantly. For now, we can only estimate where the epitope of particular clones are based on the observed cross-reactivity and knowledge of mutations. When a clone is cross reactive against the different variants, it is likely to bind an epitope that is not affected by mutations occurring across the different variants. In contrast, when a clone binds only to e.g. the WT variant and not to the other variants, it is likely that the epitope of this clone is mutated across the different variants. In response to the reviewer, we extended our discussion (lines 224-232) with a few sentences about our “limited knowledge” of the epitopes, as described below.

“Ideally, we would know the exact epitopes of all anti-spike IgG1 clones we detect in our affinity pull-downs, but this would be very labor intensive and likely require recombinant production of each of the clones. However, we can speculate about the epitopes of each clone based on the (lack of) cross-reactivity and knowledge of specific mutations occurring in the different VOCs. Clearly, when a clone displays high cross-reactivity it is not affected by the mutations and may thus bind outside the regions affected by the mutations. Conversely, when a clone does not bind to one or two spike variants but does bind to the others this provide circumstantial evidence that this mutation may be part of the epitope.”

- The authors claimed that their study can “aid in selecting antibodies that may be developed into biotherapeutics. However, since these antibodies are identified by mass spectrometry, the original B cells that produce these antibodies would not be known. Without the original B cells, how can they produce monoclonal antibodies which are exactly the same as the ones identified by the mass spectrometry? Please explain.

Previously we have shown a proof-of-concept for *de novo* sequencing IgG at the protein level (Bondt et al. Cell Systems 2021). Applying this method would allow to reproduce the antibody without knowledge about the B cells. However, for all the polyclonal Fabs detected here this would still be a very arduous and expensive task.

- In line 177-180, the authors suggest that for donor002, the majority of the abundant clones are not directed to S protein (line 177-180). However, in Figure 4B, donor002 has relatively high neutralizing antibody titer against all variants and binding antibody titer against the S protein. Please explain the apparent discrepancy between the MS-based method and the neutralization/binding antibody assays.

The reviewer is right. There is an apparent discrepancy between the MS-based method and the neutralization/ binding assays for donor 002. This discrepancy could originate from the fact that with the MS-based method we are assessing solely IgG1 binding while the neutralization and Luminex binding assays are performed on total plasma and total IgG, respectively (lines 190-194). The differences observed for this donor suggest that the immune response of this donor is governed by IgG classes beyond IgG1. We revised the text to state this more clearly, see lines 245-250.

Minor comments:

- Patient details: Were any of these patients admitted to ICU or require oxygen supplementation? Any acute complications? Was steroid or other immunomodulators given during hospitalization?

Only donor 003 was admitted to the ICU. In the ICU, this donor received oxygen supplementation: first via intubation, later via nasal prongs. This donor was admitted to the ICU before Dexamethasone or Remdesivir were given to COVID-19 patients, and was therefore only treated with antibiotics.

- Figure 1 and Supplementary Table 1. For all abbreviations, provide the full name. For example, what does IC stand for?

We agree with the reviewer’s suggestion and included the full writing of all abbreviations in Figure 1 and Supplementary Table 1.

- For supplementary Table 1, it would be useful for readers to outline the definition of the WHO scores.

In response, we included the definitions in the description of Supplementary Table 1.

Reviewer #3 (Remarks to the Author):

In the manuscript entitled “IgG1 responses following SARS-CoV-2 infection are polyclonal and highly personalized, whereby each donor and each clone displays a distinct pattern of cross-reactivity against SARS-CoV-2 variants”, the authors used MS based analytical method to discover the molecular diversity and quantity of IgG1 from 8 donors after their infection of SARS-CoV-2 and variants. The topic is important to be investigated, and some comments are listed in the following:

1. Cross-reactivity of IgG1 from donors infected by one specific strain to other variants was well discussed and investigated in this study; however, only half of the donors got sequence-confirmed results for the infected strains. This should be indicated when performing the results, for example, on page 9 and line 258~267.

We thank the reviewer for this comment but like to respond that we already mentioned that not all donors were sequence confirmed in the method section on page 12, lines 287-292.

2. As different spike proteins were used to enrich antibodies from donor plasma, specific mutations for each VOC on the spike protein are suggested to be mentioned in the introduction or discussion.

We have all the variant specific mutations mentioned in the method section (lines 299- 304).

3. As mentioned by the authors, other IgG subclasses might also play an important role in SARS-CoV-2 infection, such as IgG3. Therefore, the isotypes or subclasses information for antigen-enriched immunoglobulins are suggested to be profiled as supporting information for this study.

We agree with the reviewer that other IgG subclasses might also play an important role, and it is true that information on these other isotypes or subclasses can give relevant additional information. We are developing clonal profiling method towards other IgG subclasses, but we hope the reviewer appreciates this is beyond the scope of the present work. Instead, we addressed the comment in lines 253-254: *‘For future studies it would be interesting to not only focus on the IgG1s, but also look at the clonal profiles of other IgG subclasses e.g. IgG2, IgG3 and IgG4.’*

4. Line 253~255 and figure 4: if the three methods target different immunoglobulin populations, can they be discussed or compared together? I hope the experiment suggested in the third point will help to clarify this issue.

The reviewer is right in that the three methods used in Figure 4 potentially target different immunoglobulin populations, namely IgG1 (MS), all IgGs (binding; B) and whole plasma (neutralization; N). However, we hypothesized that it is still interesting to compare such evaluations. Although or MS-based method is solely IgG1 focused, this often represents the most abundant subclass of spike specific IgG. Based on that assumption the data should be comparable to the total IgG binding. When the different methods are not in agreement with each other, one could speculate about IgG classes other than IgG1 playing a substantial role in binding towards the spike protein. The same is true for the neutralizing assays; in case of disagreement one could speculate about a larger role of Igs beyond IgGs. We feel we have mentioned this openly and clearly in the manuscript in the discussion (lines 245-251).

5. Among the eight donors, participants 307 and 308 showed the least functional antibodies (figure 4B). Is there any comment?

That donor 307 and 308 show the lowest amount of spike-binding antibodies can potentially be explained by their relatively young age (both 18 years old) combined with their low severity score. We added this comment to the revised manuscript, lines 254-256.

6. Many studies have characterized B cell receptor and humoral immune system to the COVID-19. They should be included in the discussion and compared to the current study too. In addition, why the authors selected IgG1 to investigate the diversity should be introduced in the work.

Indeed, quite a few studies have characterized B cell receptor and humoral immune system responses to COVID-19. It may be that we still miss a few, but we added to the discussion (lines 210-217):

“Previous studies investigating the humoral immune responses of SARS-CoV-2 infected persons used either NGS (Schultheiß et al. Immunity 2020, Niu et al. Front Immunol 2020, Zhou et al. J autoimmune 2021) or ELISA (Guo et al. Clin Infect Dis.2020, Zhao et al. Clin Infect Dis. 2020, Lv et al. Cell Rep. 2020, Beaudoin-Bussièrès et al. mBio 2020) based methods. While NGS-approaches provide insights into specific clones at the DNA or RNA level, they do not provide direct information about the abundances of these SARS-CoV-2 specific clones in circulation. The ELISA-based assays provide information on the produced Igs after SARS-CoV-2 infection, however, such assays lack the discriminatory power to provide insights at the level of individual unique antibody clones.”

REVIEWERS' COMMENTS

Reviewer #1 (Remarks to the Author):

I am satisfied with the response of the authors and I recommend publication.

Reviewer #2 (Remarks to the Author):

The authors have provided satisfactory responses to the comments.

Reviewer #3 (Remarks to the Author):

The manuscript entitled "Dissecting cross-reactivity in polyclonal IgG1 responses against SARS-CoV-2 variants of concern" has been revised according to the comments. I would like to appreciate the authors spending time adding more explanations and comments in the revised manuscript.

REVIEWERS' COMMENTS

Reviewer #1 (Remarks to the Author):

I am satisfied with the response of the authors and I recommend publication.

We thank reviewer #1 for reviewing our manuscript and recommending publication.

Reviewer #2 (Remarks to the Author):

The authors have provided satisfactory responses to the comments.

We thank reviewer #2 for reviewing our manuscript and being satisfied by the comments.

Reviewer #3 (Remarks to the Author):

The manuscript entitled "Dissecting cross-reactivity in polyclonal IgG1 responses against SARS-CoV-2 variants of concern" has been revised according to the comments. I would like to appreciate the authors spending time adding more explanations and comments in the revised manuscript.

We thank reviewer #3 for reviewing our manuscript and the kind words.